# Psychosomatic Symptoms and Neuroticism following COVID-19: The Role of Online Aggression toward a Stigmatized Group

**DOI:** 10.3390/ijerph18168672

**Published:** 2021-08-17

**Authors:** Fei Teng, Xijing Wang, Jiaxin Shi, Zhansheng Chen, Qianying Huang, Wanrong Cheng

**Affiliations:** 1School of Psychology, South China Normal University, Guangzhou 510631, China; tengfei.scnu@gmail.com (F.T.); piper.hqy@m.scnu.edu.cn (Q.H.); 2019023004@m.scnu.edu.cn (W.C.); 2Department of Social and Behavioral Sciences, City University of Hong Kong, Hong Kong, China; 3Department of Psychology, The University of Hong Kong, Hong Kong, China; billshi@connect.hku.hk (J.S.); chenz@hku.hk (Z.C.)

**Keywords:** COVID-19, on-line aggression, neuroticism, psychosomatic symptoms, mental health

## Abstract

The present study investigated the effect of interpersonal mistreatment on the perpetrators’ mental health. We proposed that the threat of COVID-19 will increase people’s mental health problems through their on-line aggression toward stigmatized groups accused of spreading the disease and that there might be potential gender differences in such effects. We tested our predictions among a sample of U.S. residents (Study 1) and a large sample of Chinese residents living out of Hubei province (Study 2) during a heightened period of concern about COVID-19, February 2020. Specifically, we measured U.S. residents’ on-line aggressive behaviors toward Chinese people (Study 1) and Chinese non-Hubei residents’ on-line aggressive behaviors toward Hubei residents (Study 2) as well as their neuroticism (Study 1) and mental health states (Study 2). In line with our predictions, both studies showed that perceived infection of COVID-19 can induce on-line aggression toward stigmatized groups, thereby increasing people’s mental health problems. Moreover, the relationship between COVID-19 vulnerability, on-line aggression, and psychosomatic symptoms was more prominent in men than in women. These results offer insights into people’s responses toward COVID-19 and add to the understanding of people’s mental and physical health during the epidemic stage of contagious diseases.

## 1. Introduction

“The guy who tried to kick me then said, ‘I don’t want your coronavirus in my country’, before swinging another sucker punch at me, which resulted in my face exploding with blood”.

—A hate crime victim in London

The novel coronavirus (COVID-19) has now affected almost two hundred million people, causing more than three million deaths by June 2021. The outbreak of COVID-19 has had many side effects. Financial markets are crashing; the travel and entertainment industries are devastated; and the global conversation has been dominated by mass hysteria and panic. However, one of the most devastating side effects has been the stigmatization and mistreatment of people accused of spreading the epidemic, that is, Wuhan residents for people living in China and Chinese people those living in other countries. Although it is a reasonable precautionary measure to avoid close contact with potential disease carriers, the prevailing avoidance and even attack of Wuhan residents in China and Chinese people globally regardless of their health status connotes a sense of stigmatization and hatred. In fact, the interpersonal mistreatment of these stigmatized groups likely resulted in mental and physical dysfunctions beyond the impact of the actual biological agent. Previous research has demonstrated that derogation and avoidance of stigmatized groups arise as a response to infectious deceases and negatively influence people’s mental and physical well-being [1,2,3]. However, with an emphasis on the targets of mistreatment, this line of research largely neglected its effect on the perpetrators of such mistreatment or stigmatization. In the present study, we aimed to investigate the effect of interpersonal mistreatment on people’s mental health on the side of perpetrators. We proposed that the threat of COVID-19 will increase people’s mental health problems through their mistreatment of stigmatized groups.

### 1.1. Stigmatization of Hubei/Chinese Residents

Human beings possess a “Behavioral Immune System” for preventing the transmission of pathogens by promoting early detection and behavioral avoidance of people exhibiting disease-relevant cues [4,5,6]. Because of the potential costs of misses (false negatives) in identification, people tend to overgeneralize cues [7] to include those exhibiting cues that are heuristically (though perhaps falsely) associated with disease [8]. Researchers have posited that associative stigma could broaden to a city, a country, a region, or an entire ethnic group perceived to be at high risk of diseases [9,10]. For the case of COVID-19, one such cue is people’s residential identity. The outbreak of COVID-19 globally imposed great costs on people’s physical health and well-being; it had a relatively clear origin and consequently was viewed as a danger emanating from Wuhan, China. Some even call COVID-19 as the “Wuhan Virus” or “China Disease” despite the endeavor that both the WHO and the Centers for Disease Control (CDC) made to distribute accurate information to fight the stigma against people of such groups. Media reports suggest that stigma and hostile treatment related to COVID-19 emerged in many domains of everyday life, such as workplaces, schools, restaurants, and shopping centers. In the current research, we investigate the possible influences of COVID-19 on aggression toward the stigmatized group. Moreover, due to its high infectiousness and human-to-human transmission, to curb the epidemic of this respiratory disease, many countries have implemented traditional public health measures such as isolation and quarantine, which limit people’s opportunities for social contact. Residents of affected areas also actively limit their social participation as a precautionary measure to protect themselves from the disease, especially those who perceive the disease as severe and difficult to control. Because of the isolation and decrease in social participation due to COVID-19, people rely more on the internet to gain information and interact with others. Therefore, we focused on the influence of COVID-19 on on-line aggressive behaviors toward stigmatized groups, such behaviors include verbal attack, spreading improper information, or excessive avoidance through the internet.

### 1.2. On-Line Aggression toward Stigmatized Groups

Aggression is conceptualized as behaviors intended to cause others injury or discomfort [11,12]. Generally, aggressive behaviors can take various forms including physical aggression (e.g., hitting, pushing, and kicking), verbal aggression (e.g., calling mean names in a hurtful way), social exclusion (e.g., ignoring or leaving out others on purpose), and spreading rumors (e.g., telling lies about others) [13]. Recently, on-line aggression has emerged as a new type of aggression, which involves inflicting harms on others using electronic devices such as cell phones and computers [14,15,16]. Typical online aggressive behaviors include denigration (insults and humiliation), offensive or threatening messages or calls, identity theft, exclusion, the publication of confidential information, manipulation of photographs, the recording of physical assaults that are subsequently disseminated, etc. [17,18]. Due to the ease of acting anonymously and the perception of greater distance between the aggressor and the target, online aggression may not require the same degree of moral disengagement at the individual level [19,20] and, therefore, is easier to perpetrate. Previous research on online aggression was mostly conducted in school settings and has demonstrated that victims suffer from severe mental problems such as anxiety, depression, stress, sleep problems, and suicidal thinking and behaviors in extreme cases [21,22,23,24].

There are different forms or motives of aggression [25,26,27,28]. More pertinent to the present research, people choose aggression as a self-defense response to threats [28,29]. For example, research demonstrated that women would be more likely to report perpetrating physical aggressive behaviors for self-defense against a dating partner [30]. Another study found that participants exhibited increased aggression toward the confederate in a high social threat condition [31]. Additionally, coping with a math test in a threatening intellectual environment resulted in aggressive responses from women [32]. Moreover, other researchers have posited that the propensity for aggressive behaviors may be linked to attentional biases involved in avoiding threat-relevant stimuli [33,34]. Concurring with this perspective, Buades-Rotger and Krämer [35] used an emotional word Stroop task to examine the relationship between threatening semantic information and aggressiveness, which indicated that a tendency to dwell on implicit risky cues can reflect enhanced aggression. Taken together, the above theorizing and evidence suggest that individuals in a threatening context may behave aggressively.

In so far as COVID-19 imposes great threats to physical health and well-being, it is plausible that people would respond to the threat with hostility. In fact, previous research on infectious diseases has extensively investigated people’s defensive responses toward such threats and demonstrated a predictable behavioral pattern of excessive avoidance and hostile treatment of stigmatized groups accused of imposing such threats. For example, Leary and Schreindorfer [36] argued that people will dissociate from (that is, avoid, exclude, or ostracize) those who possess shared characteristics with the stigmatized group. In terms of infectious diseases such as AIDS/HIVS that carry a significant and harsh social stigma [1,2,37], Herek and Capitanio [38], it was found that more than a third of the respondents in their survey indicated that the stigmatized group (i.e., people with AIDS) should not be allowed in society, and many indicated a strong desire to avoid physical interaction with them. Similarly, Mak et al. [39] found comparable stigmatization of AIDS, severe acute respiratory syndrome (SARS), and tuberculosis (TB), such that people reported antagonistic affections and exhibited behavioral avoidance tendencies toward social groups with these diseases. Research on the residents of Amory Garden, the first officially recognized site of the community outbreak of SARS in Hong Kong, revealed that most of them were stigmatized and, thus, rejected and mistreated in domains of work, interpersonal relationship, and schooling [3]. This line of research provides indirect evidence that people might respond defensively toward a stigmatized group accused of spreading infectious disease. Therefore, we proposed that the threat of COVID-19 will increase aggression toward people accused of spreading the disease.

### 1.3. Vulnerability and Aggression

According to protection motivation theory [40], which is built on stress appraisal theory [41], the perceived threat of and consequent responses toward a health risk depend on one’s perceived vulnerability to that risk. Specifically, when people perceive that they have high vulnerability to a risk but low efficacy to protect themselves from it, they will experience psychological threat and respond defensively. Correspondingly, previous research found that people’s strategies for avoiding people with AIDS were correlated with their beliefs about contagiousness [38]. Similarly, research on people’s responses toward Ebola revealed that perceived vulnerability to the disease positively predicted xenophobic behaviors toward outgroups (e.g., prejudice toward West Africans, support for restrictive travel policies) [42]. A national survey on 4607 Chinese citizens demonstrated that perceived severity of COVID-19 was predictive of increases in negative emotions as well as precautionary behaviors and decreases in social participation [43]. Therefore, we proposed that perceived vulnerability to COVID-19 will facilitate expressed hostility and antagonist responses, that is, on-line aggression toward the stigmatized groups.

### 1.4. Mental Problems of Perpetrators

We further proposed that on-line aggression toward the stigmatized group due to COVID-19 will increase the mental health risks of its perpetrators. Past research on school bullying suggested that aggressors reported more psychosocial adjustment problems such as higher levels of stress and depression as well as lower levels of satisfaction with life than did those not involved in aggression [44,45,46,47]. Similarly, perpetrators of intimate violence reported more mental health problems such that they perceived more stress, insecurity, and depression than those not involved in such violence [48]. Moreover, on-line aggression was found to be linked to mental disorders such as loneliness, depression, anxiety, and suicidal ideation [49,50,51]. Therefore, we proposed that perceived infection of COVID-19 will increase people’s on-line aggression toward stigmatized groups, thereby further increasing psychosomatic symptoms.

### 1.5. The Current Research

We tested our predictions among a sample of U.S. residents and a large sample of Chinese residents living out of Hubei province during a heightened period of concern about COVID-19, February 2020. Specifically, we measured U.S. residents’ on-line aggressive behaviors toward Chinese people (Study 1) and Chinese non-Hubei residents’ on-line aggressive behaviors toward Hubei residents, as well as their neuroticism (Study 1) and mental health states (i.e., anxiety and depression, Study 2). We predicted that the perceived threat of COVID-19 will positively predict people’s psychosomatic symptoms through on-line aggression toward the stigmatized group.

Moreover, previous research demonstrated that men respond more strongly to danger-connoting contextual cues than women do. For example, compared to women, men perceive a greater threat within intergroup contexts [52] and express higher levels of intergroup prejudice [53,54,55]. Moreover, Yuki and Yokota [56] found that the intergroup prejudices of men tend to be especially responsive to contextual cues connoting vulnerability such that intergroup threat increased men’s (but not women’s) discrimination toward the outgroup. Similarly, male participants demonstrated a higher level of social dominance orientation than did female participants with respect to outgroup threat priming in a laboratory experiment [57]. Moreover, chronic vulnerability to darkness and ambient darkness interacted to predict activation of danger-connoting stereotypes for both male and female perceivers; however, the effect was much stronger among men [58]. A social role perspective of the gender differences in aggression proposed that as aggressive behaviors is more prescriptive of men’s gender role, men might be more likely to use it to comply with normative expectations [59]. In fact, research on gender roles also demonstrated that men respond aggressively to cues that threaten their security about their gender status [60]. Given that COVID-19 imposes a great threat to people’s health and security, it is plausible that men will respond more aggressively than women. Therefore, we will also focus on potential gender differences in the proposed relationship and predict that the link between COVID-19 vulnerability, on-line aggression, and psychosomatic symptoms will be more prominent in men than in women.

## 2. Study 1

Study 1 aimed to test our prediction in a sample of U.S. residents. We predicted that the perceived threat of COVID-19 will positively predict U.S. residents’ neuroticism through on-line aggression toward Chinese people. Specifically, we used neuroticism as an index of mental health problems in the present study because neuroticism has long been considered as one of the personality traits most relevant to psychopathology, especially anxiety and depression [61,62]. An abundance of studies have shown that neuroticism is associated with life distress, emotional disorders, substance abuse, psychotic symptoms, and physical-tension-related symptoms [63,64,65,66]. Therefore, neuroticism is regarded as a reflection of a person’s mean levels of distress over a period of time [67].

### 2.1. Method

Participants. Seven hundred and four participants were recruited through Amazon’s Mechanical Turk for the present study (337 women) [68]. The vast majority of participants (73.7%) were White, 15.5% Black or African American, 6.7% Asian, 0.4% native Hawaiian or Pacific Islander, and 3.1% other. Participants’ ages ranged from 19 to 78 years (*M* = 42.64, *SD* = 12.72), there were gender differences in age and education. Women were older (*M* = 44.39; *SD* = 12.14) than were men (*M* = 41.04; *SD* = 13.03), *p* <0.001, Cohen’s d = 0.27, while men (*M* = 4.57, *SD* = 1.30) reported higher education levels than women (*M* = 4.33, *SD* = 1.34), *p* = 0.016, Cohen’s d = 0.18.

Procedure and Measures. After providing informed consent, participants were invited to take an online survey about “Coronavirus outbreaks and personal feelings”. They completed the following measures presented in a randomized order and then provided their demographic information (i.e., age, gender, race, and education). Participants were fully debriefed at the end of the study.

*Perceived Infection.* Participants were asked to indicate the extent to which COVID-19 was infectious with a seven-point Likert scale ranging from 1 (extremely low) to 7 (extremely high). Three items were used to assess perceived infection including “Regarding this novel coronavirus disease, how infectious do you think this disease is?”, “Regarding this novel coronavirus disease, what do you think of the death rate caused by this disease?”, “Regarding this novel coronavirus disease, what is your chance of being infected by this disease?”. Mean scores were calculated, and higher scores indicated higher levels of perceived infection. The Cronbach’s α coefficient in the present study was 0.60.

*On-line Aggression.* We used three items to measure participants’ on-line aggressive behaviors toward Chinese people because of the coronavirus diseases. Reponses were made on a seven-point scale ranging from 1 (not at all) to 7 (very much). The items were “If there is any on-line information about Chinese people’s improper behavior, I will share it with others immediately”, “I find the verbal attack on-line toward Chinese people reasonable”, and “I would remind people around me to avoid Chinese people on-line”. Mean scores were calculated, with higher scores indicating higher levels of on-line aggression. For the current study, the Cronbach’s α coefficient was 0.91.

*Neuroticism*. We assessed participants’ neuroticism as an indicator of their mental health with the four-item Neuroticism Subscale of the International Personality Item Pool (the Mini IPIP) [69]. This scale is designed to measure individuals’ inclination to experience negative affect and their capacity to maintain emotional stability. Sample items were “I have frequent mood swings” and “I get upset easily”. Reponses were made on a seven-point scale ranging from 1 (extremely uncharacteristic of me) to 7 (extremely characteristic of me). The ratings were averaged (reversed when necessary) to index neuroticism, with a higher score indicating a higher level of neuroticism. The Cronbach’s was 0.76 in the present study.

*Subjective Socioeconomic Status (SES).* Participants completed the MacArthur Scale of Subjective Socioeconomic Status [70] by viewing a 10-rung ladder that represents people’s standing in society. Higher rungs indicate higher social class. Participants were instructed to rate the position they currently stand at on the ladder (1 = lowest to 10 = highest). A higher score indicated a higher SES.

### 2.2. Results and Discussion

Table 1 presents the means and standard deviations for all the predictors and outcome variables by gender. A multivariate analysis of variance (MANOVA) demonstrated that men reported more aggressive behaviors (*M* = 2.96, *SD* = 1.89) than did women (*M* = 2.18, *SD* = 1.58), *F* (1, 702) = 34.51, *p* < 0.001, partial *η^2^* = 0.047. However, men and women did not differ on perceived infection, *F* (1, 702) = 0.83, *p* = 0.36, and neuroticism, *F* (1, 702) = 0.01, *p* = 0.92.

Table 1 also presents the zero-order correlations for all the predictors and outcome variables by gender. As anticipated, for both men and women, perceived infection was positively correlated with neuroticism and on-line aggression.

#### 2.2.1. Moderation Analyses

Using the Process macro (Model 1) [71] with 95% bias-corrected and accelerated confidence intervals and 5000 bootstrap resamples, we examined the potential interaction effect of perceived infection (mean-centered) and gender (men = 1; women = 0) on neuroticism and on-line aggression, with age, education, and SES being controlled. The results reveal a significant interaction between perceived infection and gender on neuroticism, *β* = 0.43, *SE* = 0.10, *t* = 4.13, *p* < 0.001 (see Figure 1). Simple slopes analyses indicated that, for men, perceived infection was positively associated with neuroticism, *β* = 0.43, *p* < 0.001; however, this association was not found for women, *β* = −0.0004, *p* = 0.996.

Similarly, the interaction between perceived infection and gender was also significant for on-line aggression, *β* = 0.40, *SE* = 0.11, *t* = 3.51, *p* < 0.001 (see Figure 2). Simple slopes analyses indicated that, for men, higher levels of perceived infection were associated with higher levels of aggression, *β* = 0.66, *p* < 0.001; for women, the link between perceived infection and aggression was also significant, but the magnitude was smaller, *β* = 0.26, *p* = 0.003.

#### 2.2.2. Moderated Mediation Analysis

We used Hayes’ [71] PROCESS macro (Model 7) to examine whether perceived infection and gender interacted to predict neuroticism through aggression, with age, education, and SES being controlled. Results revealed that gender significantly moderated the indirect association between perceived infection with neuroticism (*index* = 0.07, *SE* = 0.03, 95% CI [0.02, 0.14]) through aggression (see Figure 3). There was a significant indirect effect for different genders: female, *a*b* = 0.05, *SE* = 0.02, 95% CI [0.01, 0.09]; and male, *a*b* = 0.12, *SE* = 0.03, 95% CI [0.07, 0.18]. The indirect effect of perceived infection on neuroticism through aggression was stronger for men than for women.

Study 1 demonstrated that U.S. residents especially men who were more vulnerable to the disease were more likely to exhibit on-line aggressive behaviors toward Chinese people, which further increased their psychosomatic symptoms.

## 3. Study 2

Study 2 was aimed to replicate the findings of Study 1 among a large sample of Chinese non-Hubei residents. Moreover, to better capture people’s mental health states, we used the Hospital Anxiety and Depression Scale (HADS) [72] and the Patient Health Questionnaire (PHQ) [73] to measure participants’ levels of anxiety and depressive symptoms. We predicted that the perceived threat of COVID-19 will positively predict Chinese non-Hubei residents’ mental health problems through their on-line aggression toward the people of Hubei.

### 3.1. Method

Participants and Procedure. Two thousand and eight Chinese people living out of Hubei Province participated and were invited to take an online survey about “Coronavirus outbreaks and personal feelings”. They provided their responses through an online data collection platform (i.e., Wenjuanxing) and received a small monetary reward. Fifty-seven participants failed an attention check and were excluded from data analyses, leaving us a final sample of 1951 participants (1377 women, *M_age_* = 21.02, *SD* = 4.92, age range = 13–54 years). All the following measures were presented in a randomized order. After finishing all the tasks, participants provided demographic information and were fully debriefed. There were no significant gender differences in participant’s age and education levels (*ps* > 0.05).

Measures. The first author translated all scales originally written in English into Chinese, and a bilingual Psychology professor then back-translated them into English. Modifications were made until all the authors agreed that the back translation matched the original meaning of the English version.

*Perceived Infection*. Participants completed the same measures as Study 1 to indicate the perceived infection of COVID-19. All items were rated on a seven-point scale ranging from 1 (extremely low) to 7 (extremely high), with higher scores indicating higher levels of perceived infection. Cronbach’s alpha was 0.63.

*On-line Aggression.* We used the same items to measure online aggression with one modification. To closely match the situations in China, the object of aggression was replaced with “Wuhan people”. All items were rated on a seven-point scale ranging from 1 (not at all) to 7 (very much), with higher scores indicating higher levels of aggression. Cronbach’s alpha was 0.62.

*HADS.* The 14-item HADS [72] is used in clinical samples as well as in the general population and demonstrated good validity [74]. This scale consists of two subscales, with seven items measuring symptoms of anxiety (e.g., “I get a sort of frightened feeling as if something awful is about to happen”), and another seven items measuring depression symptoms (e.g., “I have lost interest in my appearance”). Participants were instructed to indicate their agreement with each of the items on a four-point scale ranging from 0 (never) to 3 (always). Ratings were averaged (reversed when necessary) for each subscale to yield separate scores for anxiety and depression, with higher scores indicating greater levels of anxiety and depression. Cronbach’s alphas for anxiety and depression were 0.74 and 0.72, respectively.

*PHQ.* To provide converging validity on the measurement of mental health, the nine-item PHQ [73] was adopted to capture participants’ depressive symptoms during the outbreak of COVID-19. Participants provided their responses on a four-point scale ranging from 1 (never) to 4 (always). Sample items were “Little interest or pleasure in doing things” and “Trouble falling or staying asleep or sleeping too much”. The ratings were summed with a possible range of 0 to 27, with higher scores indicating greater depressive symptoms. In the present sample, Cronbach’s alpha was 0.90.

*SES.* Participants completed the same measure as in Study 1 to assess their SES.

### 3.2. Results and Discussion

Table 2 presents the means and standard deviations for all the predictors and outcome variables by gender. We conducted a multivariate analysis of variance (MANOVA) to compare mean differences between men and women on all the variables. The results demonstrated that men and women differed significantly on perceived infection, *F* (1, 1949) = 34.01, *p* < 0.001, depression as captured by HADS, *F* (1, 1949) = 14.80, *p* < 0.001, and aggression, *F* (1, 1949) = 10.48, *p* = 0.001. In particular, relative to men, women scored higher on perceived infection (*M*_women_ = 5.12, *SD*_women_ = 1.25; *M*_men_ = 4.75, *SD*_men_ = 1.39), lower on depression (*M*_women_ = 4.15, *SD*_women_ = 2.96; *M*_men_ = 4.74, *SD*_men_ = 3.39), and lower on aggression (*M*_women_ = 2.63, *SD*_women_ = 1.20; *M*_men_ =2.83, *SD*_men_ = 1.33). There were no significant differences between men and women on anxiety, *F* (1, 1949) = 0.71, *p* = 0.40, and depressive symptoms as captured by the PHQ, *F* (1, 1949) = 1.11, *p* = 0.29.

Zero-order correlations for all the predictors and outcome variables are presented by gender in Table 2. As anticipated, perceived infection was positively associated with aggression and mental health indices for both men and women.

#### 3.2.1. Moderation Analyses

We performed moderation analyses using the Process macro [71] with 95% bias-corrected and accelerated confidence intervals and 5000 bootstrap resamples to examine whether perceived infection (mean-centered) and gender (men = 1; women = 0) could interact to predict aggression and outcome variables, with age and SES being controlled. The results show that the interaction between perceived infection and gender was significant for aggression, *β* = 0.17, *SE* = 0.06, *t* = 2.84, *p* = 0.005 (see Figure 4). Simple slopes analyses indicated that for men, higher levels of perceived infection were associated with higher levels of on-line aggression, *β* = 0.29, *p* < 0.001; for women, the link between perceived infection and online aggression was also significant, but the magnitude was smaller, *β* = 0.12, *p* = 0.001.

However, the interactions between perceived infection and gender were not significant for anxiety (*β* = 0.17, *SE* = 0.13, *t* = 1.33, *p* = 0.18), depression (*β* = 0.06, *SE* = 0.15, *t* = 0.39, *p* = 0.69), and depressive disorder (*β* = −0.14, *SE* = 0.24, *t* = −0.59, *p* = 0.55).

#### 3.2.2. Moderated Mediation Analyses

We used Hayes’ [71] PROCESS macro (Model 7) to examine whether perceived infection and gender interacted to predict psychosomatic disorders through on-line aggression, with age and SES being controlled. Results revealed that gender significantly moderated the indirect association between perceived infection and anxiety (*index* = 0.03, *SE* = 0.02, 95% CI [0.01, 0.07]), depression (*index* = 0.03, *SE* = 0.02, 95% CI [0.002, 0.06]), and patient health (*index* = 0.05, *SE* = 0.03, 95% CI [0.01, 0.11]) through on-line aggression (see Figure 5). Specifically, there was a positive significant indirect effect for each gender on anxiety (female, *a*b* = 0.02, *SE* = 0.01, 95% CI [0.01, 0.04]; male, *a*b* = 0.06, *SE* = 0.02, 95% CI [0.02, 0.10]), depression (female, *a*b* = 0.02, *SE* = 0.01, 95% CI [0.004, 0.04]; male, *a*b* = 0.04, *SE* = 0.02, 95% CI [0.01, 0.09]), and patient health (female, *a*b* = 0.03, *SE* = 0.02, 95% CI [0.01, 0.07]; male, *a*b* = 0.08, *SE* = 0.03, 95% CI [0.02, 0.16]), suggesting that the indirect effects of perceived infection on psychosomatic disorders via aggression were stronger in men and weaker in women (see Table 3).

Study 2 replicated the findings of Study 1 and showed that Chinese non-Hubei residents who were more vulnerable to the disease were more inclined to exhibit on-line aggressive behaviors toward Hubei residents, which further increased their affective disorders, and this relationship was more prominent in men than in women.

## 4. Discussion

The present research demonstrated that perceived threat of COVID-19 positively predicted people’s on-line aggressive behaviors toward stigmatized groups (i.e., Hubei residents for Chinese and Chinese people for U.S. residents). Furthermore, the hostile mistreatment of stigmatized groups contributed to people’s psychosomatic symptoms both for Chinese non-Hubei residents and American people. Consistent with previous studies, significant gender differences emerged for the proposed relationship, such that the link between COVID-19 vulnerability, on-line aggression, and psychosomatic symptoms was much more prominent in men than in women.

Recent research on the psychological impacts of COVID-19 mainly focused on perceived vulnerability and coping efficacy toward the disease and demonstrated that, the more vulnerable people felt, the more mental illness they would experience; whereas, having confidence in taking measures to protect oneself or fight against the disease was associated with a lower risk of mental health problems [75,76,77]. These findings are in line with past studies that found that individuals’ negative appraisals about the incident (e.g., perceived risk, perceived threat, etc.) were related to more mental health problems during the outbreaks of severe acute respiratory syndrome (SARS) and Ebola [78,79,80,81]. However, relatively scant research has examined whether people’s (mal) adaptive coping strategies might influence their mental health. The present research, therefore, extended this line of research by demonstrating that people overreacted to infectious diseases by engaging in excessively avoidant and blatantly hostile behaviors toward stigmatized groups regardless of their COVID-19 status. Moreover, the stigmatization of and hostile response toward accused groups can lead to mental illness among its perpetrators, suggesting that people’s maladaptive coping strategies toward infectious disease might cause additionally deleterious effects to their mental health beyond the impacts of the biological agent [82]. Our research, thus, adds to the understanding of people’s mental and physical health during the epidemic stage of contagious diseases.

During the COVID-19 pandemic, misinformation about the disease proliferated on social media, which further exaggerated stigmatization of and hostile responses toward targeted groups accused of spreading this disease [83,84]. In addition, recent research on 1600 U.S. residents demonstrated that people share false claims partly because they fail to think sufficiently, and a reminder to check the accuracy of the message could increase the level of truth discernment in sharing intentions [85]. Importantly, the present research demonstrated that people, especially those who felt vulnerable to the COVID-19 pandemic, might deliberately share negative and often misleading information of stigmatized groups, which further might negatively influence their mental health.

Previous research on the associative stigma of infectious disease has cumulated evidence that threats to physical safety can cause the stigmatization of target groups perceived to pose such threats. Such stigmatization often manifests as excessive avoidance such as withholding social contacts, supporting restrictive policies, or rejecting entrance to public areas or usage of public facilities [3,37,86,87]. The present research put forward this line of research by showing that stigmatization of targeted groups can become extremely hostile, and therefore, may cause further detrimental effects to the mental and physical health of people perceived to possess the target group identity.

Due to social isolation and quarantine, we focused exclusively on on-line aggressive behaviors as an index of the hostile mistreatment of stigmatized groups. Previous research demonstrated high correlation between real-life aggressive behaviors and on-line aggression and demonstrated that people who were perpetrators/victims of on-line aggressive behaviors were more likely to be perpetrators/victims of real-life aggression [51,88]. Moreover, on-line aggression and real-life aggression can both cause detrimental effects to people’s mental and physical health [47,48]. Nonetheless, future research could benefit from investigating people’s real-life aggressive behaviors toward stigmatized groups as well as its potential influences on mental and physical health for both perpetrators and victims.

Although there are established questionnaires to assess online aggressive behaviors [49,89,90], these scales were designed to capture aggression in different contexts (e.g., school settings). As a result, in the current study, we created our own items to measure on-line aggression during COVID-19 using a typical behavioral index based on previous findings [89,91]. Given that Study 2 was a large-scale nationwide survey (*N* = 2008), we had to keep our assessment brief and concise. Future studies could continue to develop more systematic scales to measure on-line aggression in the context of facing and dealing with pandemic. In addition, we used people’s neuroticism as an index of mental health problems in Study 1 since neuroticism has long been considered as one of the personality traits most relevant to psychopathology, especially anxiety and depression [61,62] and it is also regarded as a reflection of a person’s mean levels of distress over a period of time [64]. Nevertheless, neuroticism can only be considered as an indirect measure of mental health problems. This issue was partially solved by Study 2, in which well-established scales (i.e., HADS and PHQ) with good validity in both clinical samples and the general population [74] were used to measure symptoms of anxiety and depression.

Studies 1 and 2 converged to demonstrate that men behaved more aggressively toward stigmatized groups due to the perceived threat of the epidemic. This result is also in line with prior findings on gender differences in the use of violent behaviors when the perpetrator feels threatened. However, the sample sizes of men and women in Study 2 were not quite balanced; therefore, the interpretation of the gender differences in people’s responses toward the COVID-19 threat should be treated with caution. Future examination should be carried out to further investigate whether men and women will respond differently toward environmental threats with a more gender-balanced sample.

The risk–resilience model proposes that risk and adversity increase the propensity of undesirable outcomes; individuals who have sufficient assets to offset the negative influence of the risk could overturn undesirable outcomes, thus showing resilience [92]. Previous research demonstrated that protection from diseases can reduce stigmatization of accused groups. For example, a study found that disease protection (vaccination and hand washing) attenuates the relationship between concerns about disease and prejudice against outgroups. Moreover, self-control as a type of psychological resources can buffer the effect of perceived vulnerability toward an infectious disease on anxiety [93]. Therefore, it warrants further investigation to test whether antivirus vaccination and other precautional measures (e.g., hand washing, wearing mask) can reduce people’s aggressive behaviors toward stigmatized groups and provide additional benefits to people’s mental health.

## 5. Conclusions

The present study investigated the effect of interpersonal mistreatment on the perpetrators’ mental health. Specifically, we measured U.S. residents’ on-line aggressive behaviors toward Chinese people (Study 1) and Chinese non-Hubei residents’ on-line aggressive behaviors toward Hubei residents (Study 2) as well as their neuroticism (Study 1) and mental health states (Study 2). In line with our predictions, two studies showed that perceived infection of COVID-19 can induce on-line aggression toward stigmatized groups, thereby increasing people’s neuroticism and mental health problems. Moreover, the relationship between COVID-19 vulnerability, on-line aggression, and psychosomatic symptoms was more prominent in men than in women.

## Figures and Tables

**Figure 1 ijerph-18-08672-f001:**
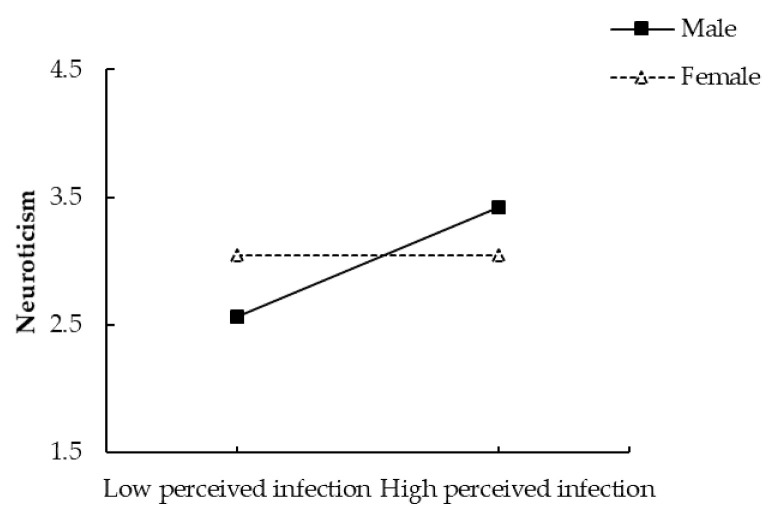
Gender as a moderator for the relationship between perceived infection and neuroticism.

**Figure 2 ijerph-18-08672-f002:**
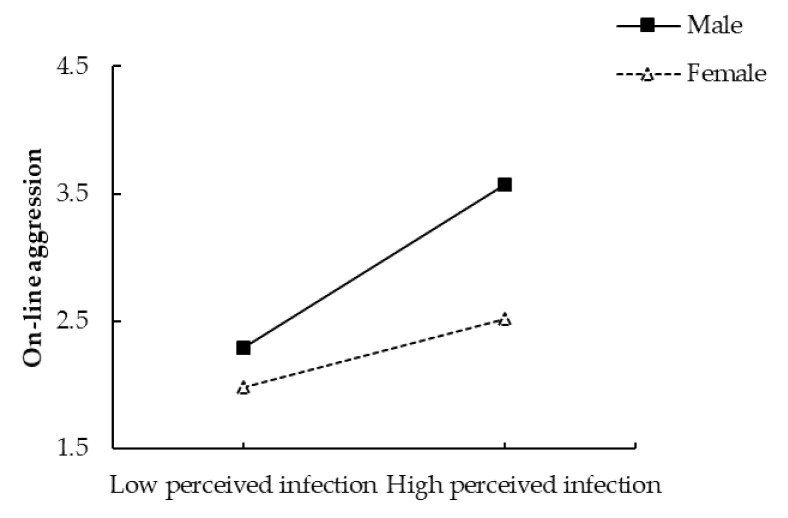
Gender as a moderator for the relationship between perceived infection and on-line aggression.

**Figure 3 ijerph-18-08672-f003:**
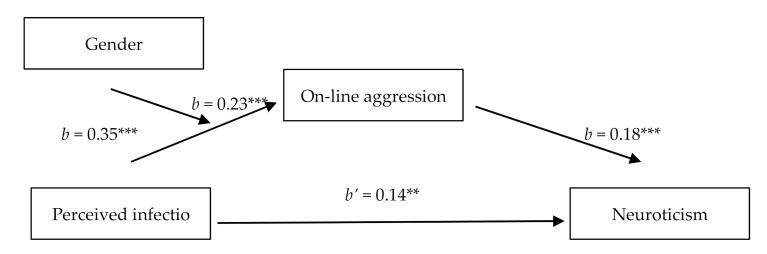
Mediation model in Study 1. ** *p* < 0.01. *** *p* < 0.001.

**Figure 4 ijerph-18-08672-f004:**
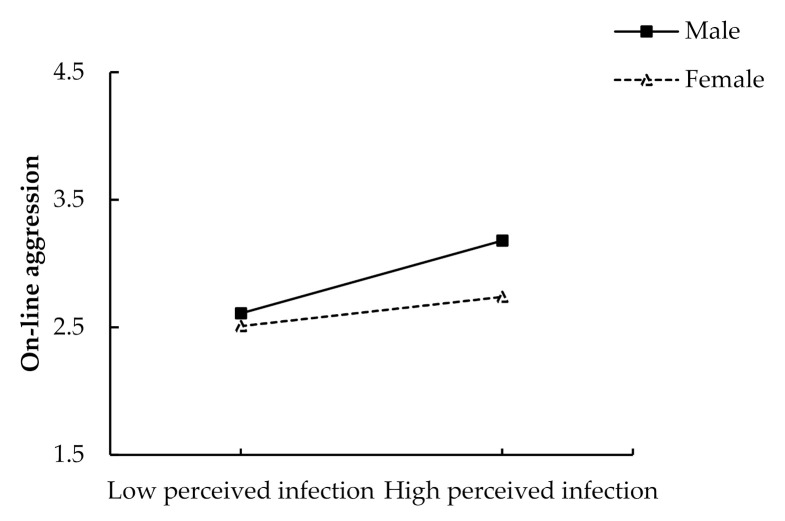
Gender as a moderator for the relationship between perceived infection and online-aggression.

**Figure 5 ijerph-18-08672-f005:**
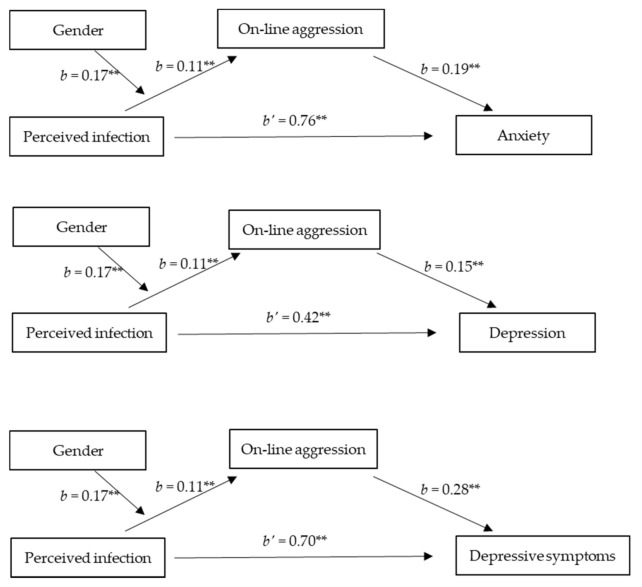
Moderated mediation models in Study 2. ** *p* < 0.01.

**Table 1 ijerph-18-08672-t001:** Descriptive statistics and correlations among variables.

							Male		Female	
	1	2	3	4	5	6	(*n* = 367)		(*n* = 337)	
							*M*	*SD*	*M*	*SD*
1. Perceived infection		0.52 ***	0.34 ***	−0.16 ***	0.38 ***	0.20 ***	4.45	1.22	4.53	1.07
2. Neuroticism	−0.004	1	0.38 ***	−0.28 ***	0.52 ***	0.27 ***	3.02	1.40	3.01	1.46
3. On-line aggression	0.20 ***	0.12 *	1	−0.25 ***	0.14 **	0.02	2.96	1.89	2.18	1.58
4. Age	−0.03	−0.27 ***	−0.12 *	1	−0.16 **	−0.02	41.04	13.03	44.39	12.14
5. SES	0.12 **	−0.14 **	0.34 ***	−0.002	1	0.47 ***	5.62	2.11	5.29	1.86
6. Education	−0.05	0.02	0.05	−0.04	0.37 ***	1	4.57	1.30	4.33	1.34

Note. Correlations for females are reported below the diagonal of the correlation matrix, while for men, they are above. * *p* < 0.05; ** *p* < 0.005; *** *p* < 0.001.

**Table 2 ijerph-18-08672-t002:** Descriptive statistics and correlations among variables.

	1	2	3	4	5	6	7	Male (*n* = 574)		Female (*n* = 1377)	
								*M*	*SD*	*M*	*SD*
1. Perceived infection	–	0.33 ***	0.18 ***	0.15 ***	0.23 ***	0.07	−0.12 **	4.75	1.39	5.12	1.25
2. Anxiety	0.26 ***	–	0.70 ***	0.54 ***	0.17 ***	0.03	−0.16 ***	4.28	3.90	4.40	2.75
3. Depression	0.16 ***	0.63 ***	–	0.59 ***	0.08 *	−0.004	−0.23 ***	4.74	3.39	4.15	2.96
4. Depressive symptoms	0.17 ***	0.55 ***	0.62 ***	–	0.15 ***	0.01	−0.17 ***	5.84	5.41	5.57	4.88
5. On-line aggression	0.10 ***	0.10 ***	0.08 **	0.06 *	–	−0.02	−0.06	2.83	1.33	2.63	1.20
6. Age	0.01	0.04	0.04	−0.03	0.004	–	0.09 *	21.29	5.50	20.90	4.65
7. SES	−0.06 *	−0.07 **	−0.1 ***	−0.09 **	−0.03	0.10 ***	–	4.59	1.77	4.64	1.56

Note. Correlations for females are reported below the diagonal of the correlation matrix, while for men, they are shown above * *p* < 0.05. ** *p* < 0.005. *** *p* < 0.001.

**Table 3 ijerph-18-08672-t003:** The moderated mediation effect of perceived infection on anxiety, depression, and depressive disorder.

Gender	Anxiety				Depression				Depressive Disorder			
	Effect	Boot SE	Boot LLCI	Boot ULCI	Effect	Boot SE	Boot LLCI	Boot ULCI	Effect	Boot SE	Boot LLCI	Boot ULCI
Female	0.02	0.01	0.01	0.04	0.02	0.01	0.004	0.04	0.03	0.02	0.01	0.07
Male	0.06	0.02	0.02	0.10	0.04	0.02	0.01	0.09	0.08	0.03	0.02	0.16

## Data Availability

The data presented in this study is fully available on request from the corresponding author.

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
