# Peer review of "Psychosomatic Symptoms and Neuroticism following COVID-19: The Role of Online Aggression toward a Stigmatized Group"

_ijerph, 2021, doi:10.3390/ijerph18168672_

Round 1

Reviewer 1 Report

The article is well written and addresses a novel and interesting theme such as the impact of online aggression to stigmatized groups related to COVID-19 on the development of psychosomatic symptoms in perpetrators groups taking into account gender. I think that although the theoretical framework has some minor issues regarding the definition of online aggression and some parts that should be better justified

  1. the word bullying is used when the authors are not speaking about this phenomenon which have some specific characteristics that differ it from whatever specific aggression;
  2. the authors have analyzed a violent phenomenon because at the end the aggression developed by perpetrators is unjustified. So, this is not an aggressive behavior, but instead a violent behavior.
  3. Gender differences in the use of violent behaviors when the perpetrator feel threatened should be better justified;
  4. more articles focused on online aggression should be included in the introduction section) it is detailed and justified in an appropriate way the aims and hypotheses. However, I consider that design of the study and specially the way to assess the constructs of the study makes the results few valid and reliable. The main reason is because the authors have designed the main measures to assess online aggression or the perception of the infection and the data seem to suggest one of them is not a reliable measure (Cronbach´s alpha of perceived infection = .60) and no information about the validity of these measures is reflected in the study. Regarding perceived infection, I can understand the need to create a new measure because we are speaking about a new disease (although it is possible that similar questionnaires are available to assess the risk of contagious of other diseases that could be used as a base). However, I don´t know why the authors have had to create the items to assess online aggression when in fact there are many questionnaires to assess this kind of violent behavior. Only it would have been necessary to add a little clarification before the items indicating that they should be answered thinking about this kind of aggression. Furthermore, it is not justified why neuroticism have been chosen as a measure of mental health when it is really a personality trait that can be high even in mental healthy people.

In this sense, the tittle of the study need to be modified because in the study there isn’t any measure of psychosomatic symptoms, maybe some questionnaires assesses psychological symptoms but not psychosomatic. So many methodological limitations make me doubt about the validity and reliability of the results and, hence, of the study.

Author Response

  1. The word bullying is used when the authors are not speaking about this phenomenon which have some specific characteristics that differ it from whatever specific aggression.

Response: Thank you for the comment. We have now replaced the word “bullying” with more appropriate words (i.e., aggression or inflicting harms on others). For example, “Recently, on-line aggression has emerged as a new type of aggression which involves inflicting harms on others using electronic devices such as cell phones and computers.” (page 2)

  1. The authors have analysed a violent phenomenon because at the end the aggression developed by perpetrators is unjustified. So, this is not an aggressive behavior, but instead a violent behavior.

Response: Thanks for the comment. We acknowledge that violence and aggression differ, at least subtly, in their meanings. For example, violence is often referred to aggressive behaviors that are meant to inflict severe harm (e.g., injury or death; Anderson & Bushman, 2002; Bushman & Huesmann, 2010; Huesmann & Taylor, 2006). However, it is difficult to clearly distinguish these two due to their mutual inclusiveness. As a result, scholars usually use these two concepts interchangeably (e.g., Siever, 2008; Dewall, Anderson, & Bushman, 2011). In addition, although aggression can be justified in certain circumstances (e.g., self-defence), most research on aggression focused on unjustified aggressive behaviors such as aggression toward people who exclude us or even innocent people when we are angry or fear of losing control (e.g., Ren, Wesselmann, & Williams, 2018; Warbuton, Williams, & Cairns, 2006; Jiang & Chen, 2020). Moreover, although violence can take many forms such as verbal or relational, it is most often researched in the context of extreme physical aggression (e.g., Allen & Anderson, 2017; Herschovis & Barline, 2006; Schat & Kelloway, 2005). Therefore, we think it is more appropriate to describe people’s maladaptive online behaviors toward COVID-19 threat such as spreading rumors or verbal attack as aggression in the present research.

  1. Gender differences in the use of violent behaviors when the perpetrator feel threatened should be better justified.

Response: Thanks for the suggestion. We have elaborated more on when we hypothesized that gender differences may emerge on violent behaviors as a response toward COVID-19 threat and added more references to support our proposition.

We have now stated: “Moreover, previous research demonstrated that men respond more strongly to danger-connoting contextual cues than women do. For example, compared to women, men perceive greater threat within intergroup contexts (Pemberton et al., 1996), and express higher levels of intergroup prejudice (e.g., Sidanius et al., 1991; Pratto et al., 1994; Sidanius & Pratto, 1999). Moreover, Yuki and Yokota (2009) found that the intergroup prejudices of men tend to be especially responsive to contextual cues connoting vul-nerability such that intergroup threat increased men’s (but not women’s) discrimination of outgroup. Similarly, male participants demonstrated a higher level of social domi-nance orientation than female participants by outgroup threat priming in a laboratory experiment (Sugiura et al., 2017). Moreover, chronic vulnerability to darkness and am-bient darkness interacted to predict activation of danger-connoting stereotypes for both male and female perceivers, however, the effect is much stronger among men (Schaller et al., 2003). A social role perspective of the gender differences on aggression proposed that as aggressive behaviors is more prescriptive of men’s gender role, therefore, men might be more likely to use it to comply with normative expectations (Eagly & Steffen, 1986). In fact, research on gender roles also demonstrated that men respond aggres-sively to cues that threat their security about their gender status (Bosson, Vandella, Burnafor, Weaver, & Arzu Wasti, 2009). Given that COVID-19 imposes a great threat to people’s health and security, it is plausible that men will respond more aggressively than women. Therefore, we will also focus on potential gender differences on the pro-posed relationship and predict that the link between COVID-19 vulnerability, on-line aggression, and psychosomatic symptoms will be more prominent in men than in women.” (please see page 4 of our manuscript for details).

  1. More articles focused on online aggression should be included in the introduction section, it is detailed and justified in an appropriate way the aims and hypotheses. However, I consider that design of the study and specially the way to assess the constructs of the study makes the results few valid and reliable. The main reason is because the authors have designed the main measures to assess online aggression or the perception of the infection and the data seem to suggest one of them is not a reliable measure (Cronbach´s alpha of perceived infection = .60) and no information about the validity of these measures is reflected in the study. Regarding perceived infection, I can understand the need to create a new measure because we are speaking about a new disease (although it is possible that similar questionnaires are available to assess the risk of contagious of other diseases that could be used as a base). However, I don´t know why the authors have had to create the items to assess online aggression when in fact there are many questionnaires to assess this kind of violent behavior. Only it would have been necessary to add a little clarification before the items indicating that they should be answered thinking about this kind of aggression.

Response: Thank you for the comment. As advised, we have now given more space to online aggression (including empirical findings. For instance, “Recently, on-line aggression has emerged as a new type of bullying aggression which involves bullying inflicting harms on others using electronic devices such as cell phones and computers (Raskauskas & Stoltz, 2007; Slonje & Smith, 2008; Williams & Guerra, 2007). Typical online aggressive behaviors include denigration (insults and humiliation), offensive or threatening messages or calls, identity theft, exclusion, the publication of confidential information, manipulation of photographs, the recording of physical as-saults that are subsequently disseminated, etc. (Campbell & Bauman, 2018; Wright, 2017). Due to the ease of acting anonymously, and the perception of greater distance between the aggressor and the target, online aggression may not require the same de-gree of moral disengagement at the individual level (Pornari & Wood, 2010; Runions & Bak, 2015), therefore, is easier to perpetrate. Previous research on online aggression is mostly conducted in school settings and has demonstrated that victims suffer from severe mental problems such as anxiety, depression, stress, sleep problems, and suicidal thinking and behaviors in extreme cases (e.g., Cross, Lester, & Barnes, 2015; Mar-tínez-Monteagudo, Delgado, Inglés, & García-Fernández, 2019b; Jenaro, Flores, & Frías, 2017; Schenk andFremouw, 2012).” (pages 2-3)

Regarding the measure of on-line aggression, although there are established questionnaires, such as Berlin Cyberbullying-Cybervictimisation Questionnaire (BCCQ; Schultze-Krumbholz, & Scheithauer, 2009), Cyberbullying Offending Scale (Hinduja & Patchin, 2009), and scales developed for a specific study (e.g., Law, Shapka, Domene, & Gagné, 2012), these scales mostly focus on adolescents’ cyberbullying at school. As such, these scales often capture aggressive behaviours such as sending nasty or rude emails or giving silent/prank phone call (towards a specific victim). As a result, these scales do not really fit the present study. Therefore, we have created our own items to measure on-line aggression during Covid-19 using typical behavioural index based on previous findings (e.g., Law, Shapka, Domene, & Gagné, 2012; Menesini, Nocentini, & Calussi, 2011). Saying these, we have now discussed this as one of our limitations. Specifically, “Although there are established questionnaires to assess online aggressive behav-iors, (e.g., Hinduja & Patchin, 2009; Law, Shapka, Domene, & Gagné, 2012; Schultze-Krumbholz, & Scheithauer, 2009;), these scales were designed to capture aggression in different contexts (e.g., school settings). As a result, in the current study, we have created our own items to measure on-line aggression during Covid-19 using typical behavioral index based on previous findings (e.g., Law, Shapka, Domene, & Gagné, 2012; Menesini, Nocentini, & Calussi, 2011). Given that Study 2 was a large-scale nation-wide survey (N = 2008), we had to keep our assessment brief and concise. Future studies could continue to develop more systematic scales to measure on-line aggression in the context of facing and dealing with pandemic.” (page 11)

Furthermore, it is not justified why neuroticism have been chosen as a measure of mental health when it is really a personality trait that can be high even in mental healthy people. In this sense, the tittle of the study need to be modified because in the study there isn’t any measure of psychosomatic symptoms, maybe some questionnaires assesses psychological symptoms but not psychosomatic. So many methodological limitations make me doubt about the validity and reliability of the results and, hence, of the study.

Response: Thank you for the comment. Neuroticism has long been considered as one of the personality traits most relevant to psychopathology especially anxiety and depression (e.g., Costa & McCrae 1980; Lucas & Fujita 2000). Neuroticism scale (i.e., NEO PI-R) includes items largely overlapping with the symptoms of depression and anxiety, such as “I feel tense and jittery” and “I worry about things that might happen”. An abundance of studies have shown that neuroticism is associated with life distress, emotional disorders, substance abuse, psychotic symptoms, physical tension-related symptoms (e.g., Clark, Watons, & Mineka,1994; Ormel, Oldehinkel,Nolen, Vollebergh,2004; Krabbendam, Janssen, Bak, Bijl, de Graff, van OS, 2002; van Os & Jones 2001; Goodwin, Fergusson, & Horwood, 2003). Therefore, neuroticism is regarded as a vulnerability measure underpinning the risk of developing an episode of depression in response to environmental adversity and even a reflection a person’s mean levels of distress over a period of time(Ormel, Rosmalen, & Farmer, 2004). We have elaborated more on why we chose neuroticism as an index of mental health problems in Study 1: “Specifically, we used neuroticism as an index of mental health problems in the present study, because neuroticism has long been considered as one of the personality traits most relevant to psychopathology especially anxiety and depression (e.g., Costa & McCrae 1980; Lucas & Fujita 2000). An abundance of studies have shown that neuroticism is associated with life distress, emotional disorders, substance abuse, psychotic symptoms, physical tension-related symptoms (e.g., Clark, Watons, & Mineka,1994; Ormel, Oldehinkel, Nolen, & Vollebergh, 2004; van Os & Jones 2001; Goodwin, Fergusson, & Horwood, 2003). Therefore, neuroticism is regarded as a reflection a person’s mean levels of distress over a period of time (Ormel, Rosmalen, & Farmer, 2004).” (page 5)

In addition, for Study 2 (N = 2008), we have used the well-established 14-item HADS (Zigmond & Snaith, 1983) which demonstrated good validity in both clinical samples as well as general population (e.g., Bjelland et al., 2002) to measure symptoms of anxiety and depression.

In sum, we have addressed each issue raised. As a result, the paper has been strengthened and improved considerably. Once again, we appreciate all the constructive feedback!

Reviewer 2 Report

Very good article. The article is well written, accurate and  original. I appreciate the topic of the article because it is very important to examine violence during this pandemic.  in this article , authors try to investigate the levels of stigmatization and mistreatment of people accused of spreading the epidemics. They analyze the percieved threat towards the epidemic and the levels of violence and aggression both in U.S. and Chinese residents.

I really like this article and kg only questions are minor questions.

1) at page 4 in your article you say that "we further proposed that online aggression of stigmatized group  due to covid 19  will increase   mental health risks  of its perpetrators". My question is: can it be the the opposite? That people with worst mental health have higher risk to perpetrate online aggression?

2) why in the second study  the number of women is so high? I suggest to insert this discrepancy in the number of male and female partecipants in the limitations.

3) in the middle part of the discussion you say that: " news that italians were marching in the streets, accusing Chinese people of bringing the illness to their country. " when you say something "strong" like this in an article,  please cite your   references. Prefereably italian or english  journals saying this or delete  this part. Unless you cite your references, this can be perceived as offensive.

I have nothing more to add. It is a good article. Thank you!

Author Response

  1. At page 4 in your article you say that "we further proposed that online aggression of stigmatized group due to covid 19 will increase mental health risks of its perpetrators". My question is: can it be the opposite? That people with worst mental health have higher risk to perpetrate online aggression?

Response: Thanks for your comment. Indeed, prior studies repeatedly demonstrated a positive association between depression and general aggression as well as spousal violence and homicide (e.g., Belfrage & Ryinbg, 2004; Painuly, Grover, Gupta, & Mattoo, 2011; Roland, 2002; Weiss & Catron, 1994). We agree with you that it is possible that people with mental health problems are more prone to perpetrate aggression. To rule out this possibility, we have done additional analysis. In particular, we have tested the mediation model on American participants (Study 1) with perceived infection as predictor, online aggression as outcome variable, and neuroticism as mediator. The results showed that the mediation effect was not significant, b = -0.139, 95%CI= [-0.35, 0.06]. We have also tested the mediation model on Chinese participants (Study 2) with perceived infection as predictor, online aggression as outcome variable, and depression and anxiety as mediators. Consistent with the results of Study 1, the mediation effect of anxiety was shaky, b = 0.008, 95% CI= [0.001, 0.014], and the mediation effect of depression was not significant, b = 0.003, 95% CI= [-0.005, 0.014].

  1. Why in the second study the number of women is so high? I suggest to insert this discrepancy in the number of male and female participants in the limitations.

Response: Thank you for pointing this out. As suggested, we have now discussed the issue in the limitation part. In particular, “Studies 1 and 2 converged to demonstrate that men behaved more aggressively toward stigmatized group due to the perceived threat of the epidemic. This result is also in line with prior findings on gender differences in the use of violent behaviors when the perpetrator feels threatened. However, the sample sizes of men and women in Study 2 were not quite balanced, therefore, the interpretation of the gender differences on people’s responses toward COVID-19 threat should be treated with caution. Future examination should be carried out to further investigate whether men and women will respond differently toward environmental threat with more gen-der-balanced sample.” (page 11)

  1. In the middle part of the discussion you say that: "news that Italians were marching in the streets, accusing Chinese people of bringing the illness to their country. " when you say something "strong" like this in an article,  please cite your   references. Preferably Italian or English  journals saying this or delete  this part. Unless you cite your references, this can be perceived as offensive.

Response: Thanks for the suggestion. We apologize for the strong language. We have now tuned the tone down. Specifically, we have now stated: “During the COVID-19 pandemic, misinformation about the disease has proliferated on social media which further exaggerated stigmatization of and hostile responses to-ward targeted groups accused of spreading this disease (Frenkel et al., 2020; Russonello, 2020).” (page 10)

In sum, we have addressed each issue raised. As a result, the paper has been strengthened and improved considerably. Once again, we appreciate all the constructive feedback!

Round 2

Reviewer 1 Report

I think that authors have improved some parts of the text regarding reviewer´s previous comments. Other parts in which some issues were indicated have not been changed although authors have tried to justify their pertinence or include them as a limitation. 

In general, I think that the manuscript is better now but I´m still worry about the tittle (I think that this work did not assess psychosomatic symptoms anyway) and the use of neuroticism as an indicator of mental health. Although authors justify that neuroticism is very related to mental health problems according to classical studies and theories of personality (I agree with this, being a risk factor, in general) this is a stable trait in time. People tend to be more or less neurotic and this is difficult to change even with psychological intervention. I think that authors should have choosen other indicators susceptible to change to assess mental health (life satisfaction, psychological adjustment, stress, anxiety, positive and negative affect....etc.) because in fact they are assessing a process that are not stable in time such as a specific type of aggression related to a specific and new phenomena as it is COVID. 

Author Response

Thank you for this point. As a result, we have modified our title and manuscript throughout. For example: “Psychosomatic Symptoms and Neuroticism following COVID-19: The role of online-aggression of stigmatized group” (title); “Specifically, we measured U.S. residents’ on-line aggressive behaviors toward Chinese people (Study 1) and Chinese non-Hubei residents’ on-line aggressive behaviors toward Hubei residents (Study 2) as well as their neuroticism (Study 1) and mental health states (Study 2).” (abstract)

In addition, we have acknowledged this point as our limitation: “we used people’s neuroticism as an index of mental health problems in Study 1 since neuroticism has long been considered as one of the personality traits most relevant to psychopathology, especially anxiety and depression (e.g., Costa & McCrae 1980; Lucas & Fujita 2000), and it is also regarded as a reflection of a person’s mean levels of distress over a period of time (Ormel et al., 2004). Still, neuroticism can only be considered as an indirect measure of mental health problems. This issue has been partially solved by Study 2, in which well-established scales (i.e., HADS and PHQ) with good validity in both clinical samples and the general population (e.g., Bjelland et al., 2002) have been used to measure symptoms of anxiety and depression.” (page 11)

Thank you again for this advice.